# Overexpression of *CDCA2* in Diffuse Large B-Cell Lymphoma Promotes Cell Proliferation and Bortezomib Sensitivity

**DOI:** 10.3390/ijms26125596

**Published:** 2025-06-11

**Authors:** Hanne Due, Asta Brogaard, Issa Ismail Issa, Maja Zimmer Jakobsen, Cathrine Sylvester, Anne Krogh Nøhr, Louiza Bohn Thomsen, Martin Kristian Thomsen, Rasmus Froberg Brøndum, Karen Dybkær

**Affiliations:** 1Department of Hematology, Clinical Cancer Research Center, Aalborg University Hospital, 9000 Aalborg, Denmark; asta.brogaard@rn.dk (A.B.); i.issa@rn.dk (I.I.I.); mzj@rn.dk (M.Z.J.);; 2Department of Clinical Medicine, Aalborg University, 9260 Gistrup, Denmark; 3Center for Clinical Data Science, Aalborg University Hospital, and Aalborg University, 9260 Gistrup, Denmark; anne.noehr@rn.dk (A.K.N.); rfb@rn.dk (R.F.B.); 4Department of Health Science and Technology, Aalborg University, 9260 Gistrup, Denmark; lbt@hst.aau.dk; 5Department of Biomedicine, Aarhus University, 8000 Aarhus, Denmark; mkt@biomed.au.dk

**Keywords:** diffuse large B-cell lymphoma, DLBCL, bortezomib, RB-CHOP, *CDCA2*, treatment response, CRISPR

## Abstract

Numerous clinical trials have attempted to improve first-line R-CHOP treatment of diffuse large B-cell lymphoma (DLBCL) through the addition or substitution of drugs. The REMoDL-B trial, testing the addition of bortezomib (RB-CHOP), revealed that ABC and molecular high-grade DLBCL patients benefit from bortezomib. The aim of this study was to achieve a better understanding of the bortezomib response in DLBCL through a functional investigation of clinically identified markers. A retrospective analysis of transcriptional and clinical data from the REMoDL-B trial was conducted to identify genes associated with bortezomib response, identifying *CDCA2*. DLBCL patients with high expression of *CDCA2* had a superior survival outcome when treated with RB-CHOP in comparison to R-CHOP, whereas no difference in outcome was observed for patients with low *CDCA2*. Moreover, *CDCA2* was found to be overexpressed in DLBCL compared to non-malignant tissue, and to have higher levels in GCB and MYC/BCL2 double-expressor patients. Functional in vitro and in vivo studies revealed that knockout of *CDCA2* decreased DLBCL cell proliferation and a bortezomib dose–response analysis showed less sensitivity in *CDCA2* knockout cells compared to control cells. This study shows that DLBCL patients with high *CDCA2* expression benefitted from the addition of bortezomib to R-CHOP and functional studies documented a direct impact of CDCA2 on the bortezomib response in DLBCL cells.

## 1. Introduction

Molecular heterogeneity is characteristic of a diffuse large B-cell lymphoma (DLBCL), causing differences in clinical outcomes depending on the cytogenetic, genetic, and transcriptional profile [1,2,3,4,5]. The initial discovery of molecular heterogeneity arose from gene expression profiling (GEP) and showed that DLBCL can be divided into cell of origin subgroups, with expression profiles resembling normal germinal-center B-cells and in vitro-activated B-cells, defined as GCB- and ABC-DLBCL, respectively. These molecular subgroups display distinct pathogenesis and clinical outcomes, with inferior prognosis in ABC-DLBCL patients [1]. In addition, DLBCL can be subclassified based on translocations of *MYC*, *BCL2*, and/or *BCL6*, defining high-grade B-cell lymphoma, also referred to as double-hit and triple-hit lymphoma [6]. GEP-based approaches have also been applied to identify MYC-driven subgroups of DLBCL, including the double-hit signature and molecular high-grade (MHG), both of which partially overlap with double- and triple-hit lymphomas [7,8]. More recently, comprehensive investigations of the mutational landscape of DLBCL have led to the discovery of several genetic subclasses [3,4,9,10].

Although the standard treatment with R-CHOP (rituximab, cyclophosphamide, doxorubicin, vincristine, prednisone) cures up to 60% of patients, recurrent and progressive diseases constitute a significant clinical challenge, where most relapsed/refractory patients die [11,12]. Multiple clinical trials have tested the addition of novel agents targeting aberrant intracellular pathways of the malignant B-cells, most with limited effect [13,14,15,16]. The REMoDL-B trial tested R-CHOP vs. R-CHOP + bortezomib (RB-CHOP) in newly diagnosed DLBCL patients stratified by cell of origin [15,16] and documented a significantly improved 5-year overall and progression-free survival for ABC- and MHG-DLBCL patients treated with bortezomib in combination with R-CHOP [16].

To achieve a better understanding of the molecular determinants of the bortezomib response we re-analyzed transcriptional and clinical data of the REMoDL-B trial and described for the first-time the aberrant expression and prognostic significance of *CDCA2* in DLBCL. Through knockout of *CDCA2*, we showed the functional impact of CDCA2 on DLBCL cell proliferation, cell cycle progression, and response to proteasome inhibitor treatment in single and combination drug studies.

## 2. Results

### 2.1. CDCA2 Is Upregulated and Associated with Bortezomib Response in DLBCL

To identify genes displaying prognostic impact in interaction with bortezomib, the REMoDL-B data (Appendix A) was analyzed by multiple Cox proportional hazards regressions. The analysis tested all expressed genes with overall survival as the outcome and included an interaction term between gene expression and the treatment arm. Identified genes were involved in a variety of biological processes, including cell cycle, transcription, and Wnt signaling among others (Appendix A). One of the strongest interactions to bortezomib response was found in *CDCA2*, which displayed a different effect on the outcome in the R-CHOP and RB-CHOP treatment arm (*p* = 0.08). Cell Division Cycle Associated 2 (CDCA2) is a cell cycle related protein, which has been documented to be upregulated in other types of cancer [17,18,19,20]. Therefore, *CDCA2* mRNA expression was examined in non-malignant and DLBCL samples, which showed higher levels of *CDCA2* in DLBCL compared to normal tissue specimens (*p* ≤ 0.001) (Figure 1A). Consistently, the *CDCA2* expression level was significantly higher in DLBCL cell lines than in normal B-cell compartments (*p* ≤ 0.001). *CDCA2* expression was not associated with clinical factors including disease stage and international prognostic index (IPI) (Appendix A); however, GCB-DLBCL patients displayed increased *CDCA2* expression in comparison to ABC and unclassified patients (*p* ≤ 0.05) (Figure 1B). Moreover, higher *CDCA2* levels were observed in MYC/BCL2 double-expressors (*p* ≤ 0.01) (Figure 1C), constituting of 36% GCB subclassed patients. The difference in *CDCA2* mRNA levels between ABC and GCB-DLBCL patients prompted us to examine its expression in refined B-cell subsets including normal B-cell differentiation subsets and B-cell-associated gene signature (BAGS) classification of DLBCL patients. A higher expression of *CDCA2* was observed in normal centroblasts and centrocytes compared to naïve, memory, and plasmablast B-cells from healthy tonsils (*p* ≤ 0.01), and likewise, the BAGS-classified centrocyte and centroblast subgroups of DLBCL tumors [2] displayed higher *CDCA2* levels than the other subtypes, although they were not statistically significant (Figure 1D, Appendix A).

A Kaplan–Meier survival analysis revealed a superior outcome in DLBCL patients with a high *CDCA2* expression when treated with RB-CHOP in comparison to R-CHOP (*p* = 0.012), whereas no difference in survival was observed for patients with low expression of *CDCA2* (*p* = 0.095) (Figure 2A,B). In multivariate Cox proportional hazards regression analyses, *CDCA2* displayed prognostic impact independently of IPI and ABC/GCB in R-CHOP treated patients, whereas no prognostic significance was observed in RB-CHOP treated patients (Figure 2C,D, Appendix A), suggesting *CDCA2* as a marker of a bortezomib response. Moreover, the prognostic value of CDCA2 was independent of double-expressor status (Appendix A).

### 2.2. Knockout of CDCA2 Suppresses DLBCL Cell Proliferation

To elucidate the biological function of CDCA2 in DLBCL, RNP and lentiviral-based CRISPR-mediated knockout (KO) of *CDCA2* was performed. Indel analysis showed heterogenous indel populations with knockout scores of 72 and 82 in RNP-transfected RIVA cells and monoclonal knockouts with scores >90 in lentiviral transduced OCILY7 and RIVA cells (Figure 3A, Appendix A). Protein depletion was documented by Western blotting (Figure 3A). For verification of control, the proliferation rates of scramble control (SCR) and parental wildtype cells were compared and revealed no difference (Appendix A). Knockout of *CDCA2* suppressed the proliferation of DLBCL cells significantly compared to SCR, with the most prominent effect observed in monoclonal knockout cells (Figure 3B). In addition, indel distribution in RNP-transfected *CDCA2-KO* cells changed over time with decreases in indel and knockout scores, as result of the overgrowth of cells without *CDCA2* knockouts (Appendix A), supporting slower proliferation rates of *CDCA2-KO* cells.

Transcriptional profiling of *CDCA2-KO* clones and SCR was performed to explore the pathways affected upon loss of CDCA2. Gene set enrichment analysis (GSEA) indicated that CDCA2 is closely related to the G2/M checkpoint and to targets of the E2F transcription factor which play a prominent role during the G1/S transition of the cell cycle (Appendix A). Additionally, CDCA2 is associated with the PI3K/AKT signaling pathway, which is also important for the regulation of cell cycle progression [21]. In agreement, flow cytometry analysis showed that loss of *CDCA2* affected the cell cycle transition between the G1- and S-phase, resulting in a significantly increased percentage of cells in G1 compared to the control (Figure 3C, Appendix A). Moreover, a phospho-flow analysis showed significantly less AKT Ser473 phosphorylation in *CDCA2-KO* cells compared to SCR (Figure 3D, Appendix A), documenting the direct impact of CDCA2 on the PI3K/AKT pathway.

DLBCL xenografts were established to explore effects of *CDCA2* loss in vivo. The tumor growth was significantly reduced in *CDCA2-KO* xenograft tumors in comparison to SCR tumors, resulting in longer survival time (Figure 3E,F). Indel analysis documented stable knockout of *CDCA2* in tumors of *CDCA2-KO* xenografts (Appendix A). Tumors were analyzed by immunohistochemistry (IHC), and increased expression of Ki-67 was observed in *CDCA2-KO* tumors in comparison to SCRs (Figure 3G, Appendix A).

### 2.3. CDCA2 Impacts Bortezomib Response in DLBCL Cells

The findings from the analysis of REMoDL-B data prompted us to investigate the role of CDCA2 in bortezomib response in single and combination dose–response screens. No difference in response was observed between parental wildtype and SCR (Appendix A). Compared to SCR, *CDCA2-KO* cells demonstrated significantly less sensitivity to bortezomib over a range of concentrations with a more modest response in the polyclonal *CDCA2-KO* cell populations compared to the monoclonal *CDCA2-KO* clones (Figure 4A–C). In concordance, lower apoptotic levels were observed in *CDCA2-KO* cells compared to SCR upon bortezomib exposure (Figure 4D,E, Appendix A). Moreover, *CDCA2-KO* cells were less sensitive to carfilzomib, a 2nd generation proteasome inhibitor (Appendix A), showing a general effect of CDCA2 on proteasome inhibitors. Combinatory drug screens using CHOP and B-CHOP showed that SCR cells possessing physiological levels of *CDCA2* were more sensitive to B-CHOP than CHOP, whereas *CDCA2-KO* cells displayed no difference in response to the two regimens (Figure 4F), illustrating CDCA2 to be cardinal and specifically mediating the bortezomib response.

Next, we set out to decipher the mechanism through which CDCA2 impact bortezomib response. CDCA2 did not affect the proteasomal inhibiting effect of bortezomib as no difference in chymotrypsin, trypsin, and caspase activity of the proteasome was observed upon bortezomib treatment between *CDCA2-KO* and SCR cells (Figure 5A). This suggests alternative mechanisms by which *CDCA2* modulates bortezomib response. As CDCA2 was documented to affect PI3K/AKT signaling at the transcriptional and phosphorylation levels, the impact of this pathway was examined. Applying the PI3K inhibitor, LY294002, a reduction in cell proliferation was observed in both *CDCA2-KO* and SCR cells; however, it was observed to a greater extent in SCR cells with a 25% reduction compared to 16% of *CDCA2*-KO cells (Figure 5B). Combinatory dose–response analysis of LY294002 and bortezomib revealed that inhibition of the PI3K pathway sensitizes cells to bortezomib irrespective of presence or absence of CDCA2 (Figure 5C), suggesting that CDCA2 affects the bortezomib response through mechanisms other than PI3K/AKT. The sensitizing effect of LY294002 is equal in SCR and *CDCA2-KO* cells, despite having a different reducing impact on cell proliferation, showing that bortezomib response through PI3K is not driven by proliferation.

## 3. Discussion

In this study, we demonstrate, for the first-time, the aberrant expression of *CDCA2* in DLBCL samples and the functional impact of CDCA2 in DLBCL cells. Overexpression of CDCA2 has been reported in colorectal, prostate, and liver cancer, among others [17,18,19,20], and it was reported to be positively correlated to the disease stage [18,19,20]. However, *CDCA2* expression did not correlate with stage or prognostic index in our data. Interestingly, higher levels of *CDCA2* were observed in normal germinal-center B-cells, indicating that *CDCA2* plays a role in normal B-cell development. In line, DLBCL patients of the molecular GCB cell of origin subclass displayed higher *CDCA2* expression than ABC-DLBCL patients.

The prognostic impact of *CDCA2* mRNA expression was documented, suggesting *CDCA2* expression as indicator of bortezomib response. Further investigation of additional clinical DLBCL cohorts could provide more information on the predictive biomarker potential of CDCA2, yet the REMoDL-B trial is, to our knowledge, the only comprehensive study of bortezomib treatment in DLBCL. Moreover, as DLBCL is a highly heterogeneous disease [1,2,3,4,5] it is unlikely that a single gene can predict the response to a given treatment and therefore further assessment of the biomarker potential is out of scope for this study. Our study should thus be considered useful for improved biological understanding of bortezomib response rather than as a single gene prognostic biomarker candidate.

The REMoDL-B trial showed subclass-specific responses to bortezomib addition, where the improved outcome of ABC-DLBCL is explained by the inhibition of the constitutive NF-ΚΒ signaling observed in this subclass [22]. This mechanism is unlikely to be the explanation for the treatment efficacy observed in MHG-DLBCL, which has been suggested to be attributed to *MYC*-driven proteolytic stress, sensitizing cells to bortezomib [16]. We observed higher *CDCA2* expression in DLBCL patients classified as MYC/BCL2 double-expressors, who display a more aggressive disease, as it is an adverse prognostic indicator [23,24]. It could be speculated that a high *CDCA2* expression is surrogate for these aggressive tumors, thus the improved clinical outcome observed upon bortezomib addition was a result of more extensive treatment of this adverse prognostic subgroup. However, double-expressor status was not of prognostic significance [16].

For functional examination of CDCA2, knockouts were established to obtain the greatest phenotypic effect in DLBCL cells, which endogenously express high levels of *CDCA2.* CDCA2 affected DLBCL cell proliferation with a reduction upon the loss of CDCA2. In agreement with other studies [17,25,26], we found CDCA2 to be involved in G1/S phase cell cycle transition, which has been suggested to be mediated by regulation of CCND1 [17,25,26]. In DLBCL xenograft mice, loss of CDCA2 reduced the tumor growth rate, in agreement with the slower proliferation confirmed in cell culture studies [17,25,27]. A higher Ki-67 expression was observed in *CDCA2-KO* tumors, suggesting genetic redundancy. Correlation between CDCA2 and Ki-67 expression have been examined in various cancer types, yet reporting contradictory findings [17,18,25]. CDCA2 and Ki-67 organize the mitotic chromosome periphery in a similar manner—by recruiting protein phosphatase 1 (PP1) to chromatin during the anaphase. The regulatory mechanisms of PP1 are identical for CDCA2 and Ki-67 [28], suggesting redundancy between these two genes.

The PI3K/AKT pathway plays a key role in modulating cell proliferation by regulating CCND1 and is crucial in DLBCL pathogenesis [29]. This study shows that the loss of CDCA2 reduces pAKT and inhibition of this pathway in SCR cells reduced the proliferation levels more when compared to *CDCA2-KO* cells. The link between proliferation, CDCA2, and PI3K signaling has been reported in other cancer types [17,27]. Moreover, we found that, despite this pathway influencing the bortezomib response, it is not the sole mechanism through which CDCA2 exerts its action. Inhibition of the PI3K/AKT pathway has previously been shown to confer bortezomib sensitivity in DLBCL cells [30] and while this is contradictory to our observation of reduced sensitivity upon *CDCA2-KO* which also diminished pAKT, it can be explained by PI3K/AKT signaling being regulated by several other genes than CDCA2.

In dose–response screens, we applied the CHOP treatment alone and in combination with bortezomib and documented CDCA2 to mediate the bortezomib response. In agreement with our findings from the analysis of the REMoDL-B data showing superior clinical outcomes of high-*CDCA2*-expressing patients when treated with RB-CHOP, the SCR cells possessing naturally high levels of *CDCA2* displayed greater sensitivity to B-CHOP than to CHOP. In parallel to the clinical observations, the cell viability of *CDCA2-KO* cells was equally affected by CHOP and B-CHOP treatment in accordance with the lack of difference in outcome between the two treatment arms for DLBCL patients with low *CDCA2* expressions. As CDCA2 does not affect the proteasomal activity and the PI3K/AKT pathway was disproved as sole mechanisms of action, it is reasonable to believe that the impact on bortezomib response is mediated by the high involvement of CDCA2 on proliferation. Clinically, it is the ABC and MHG molecular subclasses of DLBCL which benefit from addition of bortezomib [16], and as these subclasses constitute highly proliferative and aggressive tumors it can be assumed that high proliferation increases the susceptibility to bortezomib.

In summary, our present data are the first to demonstrate overexpression of *CDCA2* in DLBCL. Functional studies confirmed a direct link between CDCA2 and DLBCL cell proliferation and bortezomib response. Moreover, we found that DLBCL patients with high expression of *CDCA2* displayed superior clinical outcomes upon the addition of bortezomib to the standard R-CHOP treatment.

## 4. Materials and Methods

### 4.1. Clinical Cohorts

This study includes data from the REMoDL-B trial (R-CHOP, *n* = 469; RB-CHOP, *n* = 459) (Appendix A) and local data of normal lymph nodes (*n* = 6) and tonsils (*n* = 6), sorted B-cell subsets from tonsils, and diagnostic DLBCL biopsies (*n* = 86). Data summary and analysis are outlined in Appendix A.

### 4.2. CRISPR/Cas9 Knockout

An ABC- and GCB-DLBCL cell line was selected for functional studies to reflect the REMoDL-B trial (RIVA and OCILY7, respectively). Transfection and lentiviral delivery were applied for CRISPR-mediated knockout of *CDCA2* as outlined in Appendix A. To validate knockout, DNA was extracted (Qiagen, #69506, Hilden, Germany) and the knockout site was PCR-amplified. Amplicons were Sanger sequenced (Eurofins Genomics, Cologne, Germany) and analyzed for indels [31]. Western blot analyses were performed as previously described [32], using 20–40 µg protein.

### 4.3. RNA-Sequencing

RNA-sequencing of OCILY7 SCR and *CDCA2* knockout (*CDCA2-KO*) cells were performed in technical duplicates, as previously described [33]. GSEA was performed using hallmark gene sets from the Molecular Signatures Database for the human species [34]. DESeq2 v.1.36.0 [35] was used to estimate log2 fold changes in differentially expressed genes. Subsequently, gage v.2.46.1 [36] was used to identify enriched hallmark gene sets, considering direction of effect. Gene sets with less than 15 or more than 500 genes were excluded from the analysis.

### 4.4. Flow Cytometry

Analysis of cell cycle and AKT/AKT phosphorylation (Ser473, pAKT) was performed on ethanol fixed samples, whereas apoptosis was analyzed in unfixed samples using flow cytometry (SONY, SH800 Cell Sorter, Tokyo, Japan). Procedures are outlined in Appendix A.

### 4.5. MTS Assays

Proliferation and dose response screens were performed as previously described [32,37] using seeding concentration of 0.25 × 10^6^ cells/mL. Drug information outlined in Appendix A. For combinatory drug studies, rituximab was omitted to minimize complexity and risk of bias, as addition of human serum is required for rituximab to exert its cytotoxic effect through the complement system.

### 4.6. Proteasome Activity Assay

Cells (15,000) were seeded in 50 µL media, in 96-well plates, prior to the addition of 10 µL of saline or bortezomib (3.5 ng/mL). After 5 h of exposure, 60 µL enzyme-specific reagent (Cell-Based Proteasome-Glo^TM^, Promega, Singapore) was added and luminescence was measured by the Omega Fluostar plate reader (BMG LABTECH, Ortenberg, Germany). The setup included technical triplicates in biological triplicates.

### 4.7. Xenograft Mice

OCILY7 *CDCA2-KO* or SCR cells were subcutaneously injected into flanks of SCID mice. Tumor sizes were measured every 3rd-4th day and mice were sacrificed upon humane endpoint (*n_SCR_* = 6, *n_CDCA2-KO_* = 10). IHC and indel analysis was performed. Extended information provided in Appendix A.

### 4.8. Statistical Analysis

Statistical analyses of clinical data were performed with R v.4.4. Flow data was analyzed using FlowJo v. 10.10. Cell cycle phases were defined using a univariate cell cycle model with the Watson–Pragmatic statistic. Mean fluorescence intensity (MFI) was determined with a statical feature in FlowJo. Statistical analyses of in vitro and in vivo studies were performed using GraphPad v.10.0. One-way or two-way ANOVA were performed for comparison of means between samples and *p*-values were adjusted using Bonferroni multiple comparison testing. * *p* ≤ 0.05; ** *p* ≤ 0.01; *** *p* ≤ 0.001; **** *p* ≤ 0.0001.

## Figures and Tables

**Figure 1 ijms-26-05596-f001:**
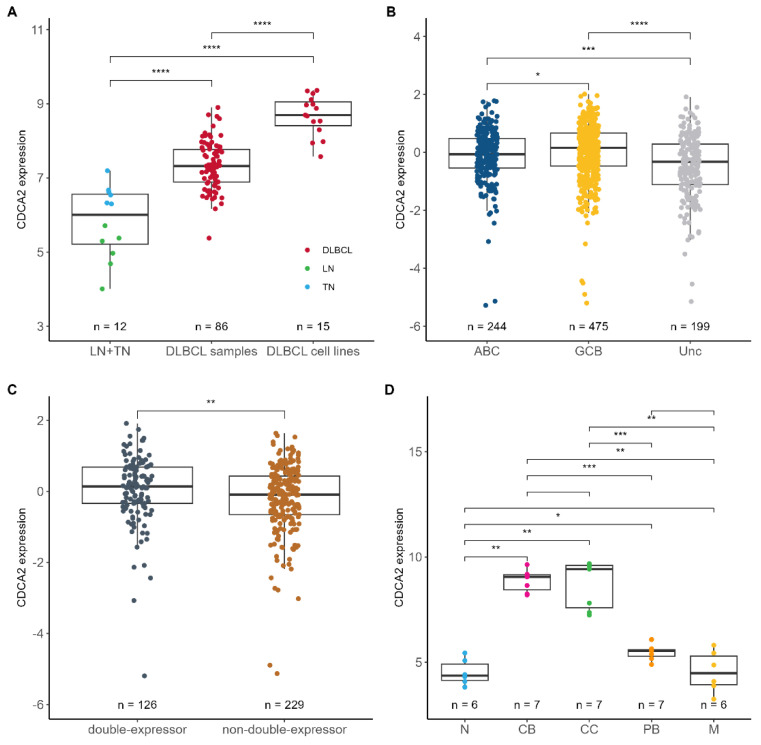
*CDCA2* expression. (**A**) *CDCA2* mRNA expression in non-malignant lymph node (LN) and tonsil (TN) tissues, diagnostic DLBCL samples, and DLBCL cell lines. (**B**) *CDCA2* expression in ABC/GCB classified patients of the REMoDL-B cohort. (**C**) *CDCA2* expression in MYC/BCL2 double- and non-double-expressor patients of the REMoDL-B cohort. (**D**) *CDCA2* expression in normal B-cell subsets FACS sorted from healthy tonsils. Wilcoxon test. * *p*  ≤  0.05; ** *p*  ≤  0.01; *** *p*  ≤  0.001; **** *p*  ≤  0.0001. ABC, activated B-cell-like; CB, centroblast; CC, centrocyte; GCB, germinal-center B-cell-like; M, memory; N, naïve; PB, plasmablast.

**Figure 2 ijms-26-05596-f002:**
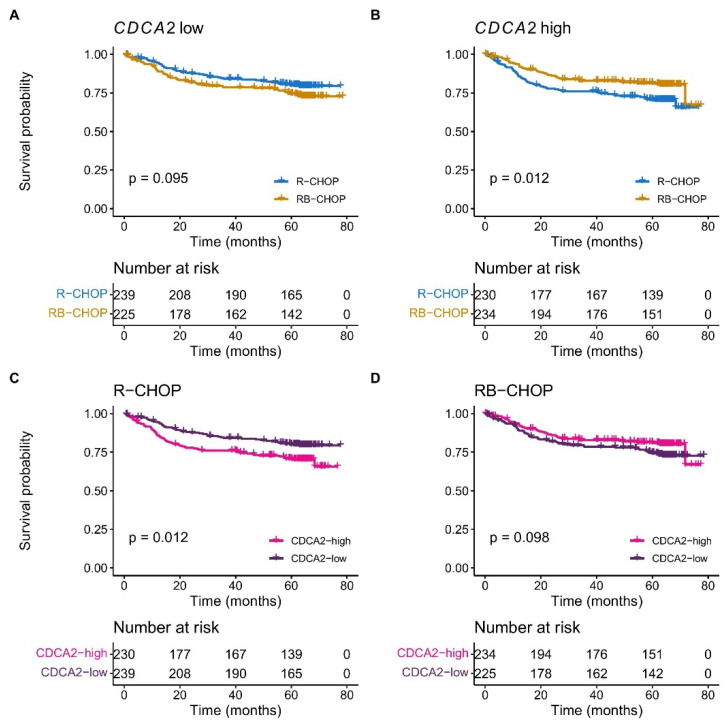
Survival analysis. Kaplan–Meier plots presenting the overall survival of DLBCL patients from the REMoDL-B cohort. Patients grouped by median split of *CDCA2* mRNA expression irrespective of treatment regimen. (**A**) Patients with low *CDCA2* expression. (**B**) Patients with high *CDCA2* expression. (**C**) Patients treated with R-CHOP. (**D**) Patients treated with RB-CHOP.

**Figure 3 ijms-26-05596-f003:**
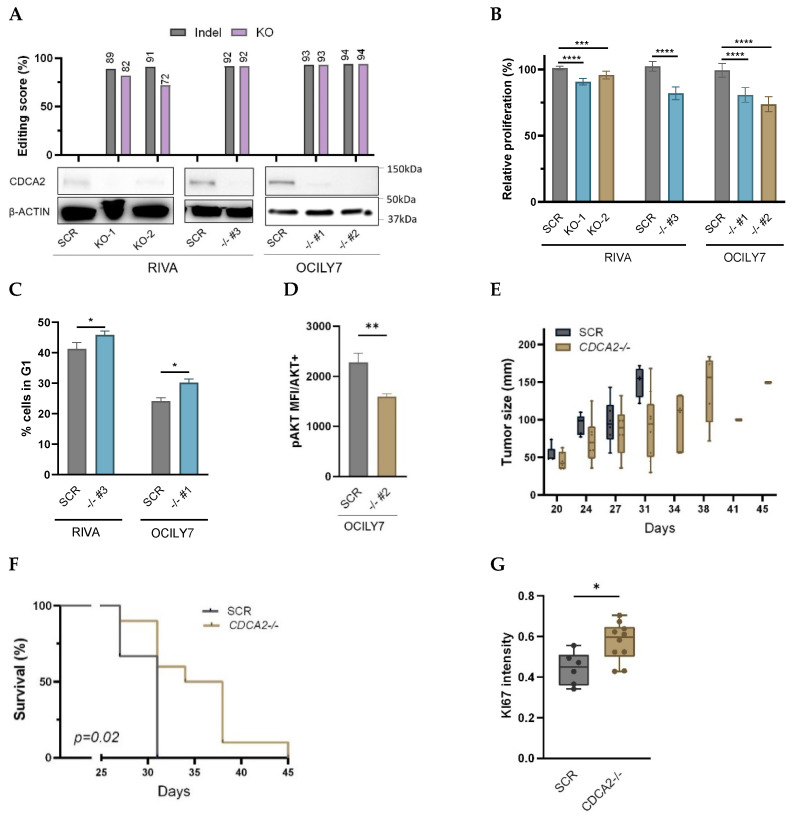
Functional investigation of CDCA2. *CDCA2* knockout cells were generated by RNP nucleofection (KO-1 and KO-2) and by lentiviral transduction (−/− #cloneID). *(***A**) Knockouts validated by indel analysis and Western blotting. (**B**) Proliferation examined by MTS presented relative to SCR. (**C**) Flow cytometry assessment of cell cycle phases. Percentages of cells in the G1-phase are depicted. (**D**) Mean fluorescence intensity (MFI) of phosphorylated AKT (pAKT, Ser473), normalized to AKT levels. (**E**) Dot plot of tumor sizes (mm) for each OCILY7 xenograft mice measured every 3rd-4th day. (**F**) Survival (%) in days from time of injection for xenografts. (**G**) Ki-67 intensity normalized to DAPI. Mean value of 10 images per sample. Values are presented as mean ± standard deviation. * *p*  ≤  0.05; ** *p*  ≤  0.01; *** *p*  ≤  0.001; **** *p*  ≤  0.0001.

**Figure 4 ijms-26-05596-f004:**
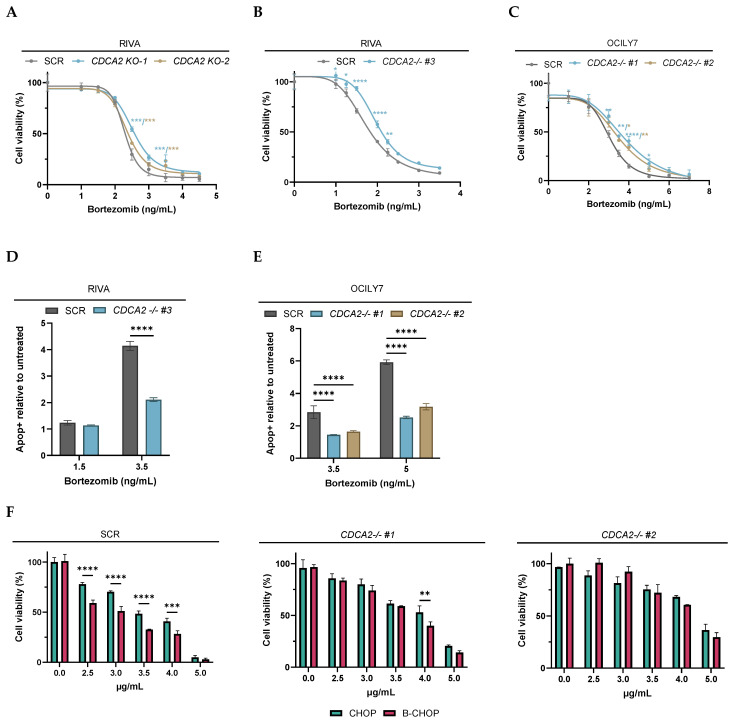
Drug response analysis of *CDCA2-KO* cells. (**A**–**C**) Bortezomib dose–response screens. Cell viability assessed by MTS and presented relative to the untreated control. (**D**,**E**) Apoptosis examined by flow cytometry upon bortezomib exposure. Apoptotic cells (Apop+) including early and late apoptotic stages presented relative to the untreated control. (**F**) CHOP and B-CHOP dose–response analysis in OCILY7. Cell viability assessed by MTS and presented relative to untreated control. KO-1 and KO-2 are heterogenous cell populations created by RNP nucleofection. −/− #cloneID represents monoclonal populations established by lentiviral transduction followed by single cell expansion. Values are presented as mean ± standard deviation. * *p*  ≤  0.05; ** *p*  ≤  0.01; *** *p*  ≤  0.001; **** *p*  ≤  0.0001.

**Figure 5 ijms-26-05596-f005:**
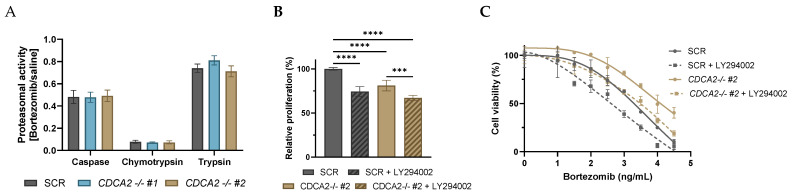
(**A**) Proteasomal activity of SCR and *CDCA2-KO* OCILY7 cells upon exposure to 3.5 ng/mL bortezomib for 5 h. Caspase, chymotrypsin, and trypsin activity presented as luminescence read-out of bortezomib cells relative to untreated cells. (**B**) Proliferation of SCR and *CDCA2-KO* OCILY7 cells examined by MTS. Cells were exposed to saline or 10 nM LY294002 for 48 h. Data presented relative to SCR in untreated condition. (**C**) Combinatory dose–response analysis of bortezomib and LY294002 (10 nM). Cell viability assessed by MTS and presented relative to untreated. −/− #cloneID represents monoclonal populations established by lentiviral transduction followed by single cell expansion. Values are presented as mean ± standard deviation. *** *p*  ≤  0.001; **** *p*  ≤  0.0001.

## Data Availability

The GEP datasets supporting the conclusions of this article are available in the National Center for Biotechnology Information Gene Expression Omnibus repository, GSE72648 (cell lines), GSE56315 (normal B-cell subsets), GSE109027 (local DLBCL cohort), and GSE117556 (REMoDL-B cohort).

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
