# Peer review of "Overexpression of *CDCA2* in Diffuse Large B-Cell Lymphoma Promotes Cell Proliferation and Bortezomib Sensitivity"

_ijms, 2025, doi:10.3390/ijms26125596_

Round 1
Reviewer 1 Report
Comments and Suggestions for Authors
The gist of this paper is that overexpression of CDCA2 in diffuse large B-cell lymphoma promotes cell proliferation and sensitivity to bortezomib.As a result, the REMoDL-B trial, which examined the addition of bortezomib (RB-CHOP), revealed the efficacy of bortezomib in patients with high-grade DLBCL. This is very interesting, but it would be good to add additional information that takes the following points into consideration.
- Why are RB-CHOP-sensitive cases often MHG-negative and GC type?
- Why is bortezomib effective in patients with high-grade DLBCL?
- Please explain CDCA2 in a more understandable way.
- Why does overexpression of CDCA2 promote cell proliferation and sensitivity to bortezomib?
- At the time of relapse, were patients receiving CAT therapy or bispecific antibody therapy?
Author Response
Comment 1: Why are RB-CHOP-sensitive cases often MHG-negative and GC type?
Response 1: Davies et al. (PMID: 36972491) and Sha et al. (PMID: 30523719) found that ABC- and MHG-classified DLBCL patients had superior clinical outcome when treated with RB-CHOP. This is described page 2, line 50-54. In our study, we show that DLBCL patients with high CDCA2 expression benefit from addition of bortezomib to the standard treatment R-CHOP. GCB-DLBCL patients have higher expression than ABC-DLBCL and higher levels is observed in double expressors (MYC/BCL2) than in non-double expressors (Figure 1B+C). However, CDCA2 mRNA expression is not an unambiguous marker distinguishing the groups. In fact, there is large spread of CDCA2 expression within each group. We chose to do multiple cox regression, revealing that CDCA2 mRNA expression display prognostic index independent of ABC/GCB and MYC/BCL2. Therefore, we did not include a theoretical speculation of why the expression is higher in GCB-DLBCL patients.
Comment 2: Why is bortezomib effective in patients with high-grade DLBCL?
Response 2: Thank you for addressing this question. To elaborate we have included following section in the discussion page 7, line 221-231:
“The REMoDL-B trial showed subclass specific responses to bortezomib addition, where the improved outcome of ABC-DLBCL is explained by inhibition of the constitutive NF-ΚΒ signaling observed in this subclass (35). This mechanism is unlikely the explanation for the treatment efficacy observed in MHG-DLBCL, which has been suggested to be attributed to MYC-driven proteolytic stress, sensitizing cells to bortezomib (16). We observed higher CDCA2 expression in DLBCL patients classified as MYC/BCL2 double-expressors, which display more aggressive disease and is an adverse prognostic indicator (36,37). It could be speculated that high CDCA2 expression is surrogate for these aggressive tumors, thus the improved clinical outcome observed upon bortezomib addition was a result of more extensive treatment of this adverse prognostic subgroup. However, double-expressor status was not of prognostic significance (16).”
Comment 3: Please explain CDCA2 in a more understandable way.
Response 3: We agree on this point. Therefore, we have included the full name Cell Division Cycle Associated 2 (CDCA2) (page 2, line 70). Moreover, the function is described page 2, line 70 and page 8, line 235-244.
Comment 4: Why does overexpression of CDCA2 promote cell proliferation and sensitivity to bortezomib?
Response 4: To examine why CDCA2 impacts DLBCL proliferation we performed gene set enrichment analysis, which identified PI3K/AKT and cell cycle checkpoints, which we subsequently evaluated by flow cytometry. This is formulated page 4, line 125-136 and page 8, line 245-255. The impact on bortezomib is discussed page 8, line 256-270.
Comment 5: At the time of relapse, were patients receiving CAT therapy or bispecific antibody therapy?
Response 5: We only analyze diagnostic biopsies and do not have access to relapse biopsies or information on second-line treatment.
Reviewer 2 Report
Comments and Suggestions for Authors
The authors performed an informatic analysis to implicate gene(s) whose expression that may underlie responses to the bortezomib arm (RB-CHOP) of the REMoDL-B trial. Having implicated CDCA2, they then demonstrated that this is supported by studies using CDCA2 knockout (KO) in DLBCL cell lines, both in vitro and in vivo.
The manuscript would benefit from attention to certain points:
1) There are some conclusions from this study that are difficult to understand:
- In the REMoDL-B trial, it was previously concluded that ABC and molecular high-grade DLBCL patients benefited from bortezomib, i.e., their outcomes were better with RB-CHOP than with R-CHOP. This study found that DLBCL patients with high expression of CDCA2 had superior survival outcome when treated with RB-CHOP in comparison to R-CHOP, whereas no difference in outcome was observed for patients with low CDCA2. From these two observations, one would expect that CDCA2 is more highly expressed in ABC-DLBCL, at least for a subset of patients, than in GCB-DLBCL. However, it was found that CDCA2 was expressed more highly in GCB-DLBCL cases and in MYC/BCL2 double-expressor patients, which more commonly belong to the GCB group. What is the explanation for these apparent discrepancies?
- Starting on line 98 of the Results, it is stated that “CDCA2 displayed prognostic impact independently of IPI and ABC/GCB in R-CHOP treated patients whereas no prognostic significance was observed in RB-CHOP treated patients (Supplementary table 3A+B), suggesting CDCA2 as marker of bortezomib response.” Although cell line experiments showed that CDCA2-KO reduced sensitivity to bortezomib, how can CDCA2 be a marker of bortezomib response if no prognostic significance was observed in RB-CHOP treated patients? Perhaps what is needed is a change to Figure 2. Instead of comparing R-CHOP to RB-CHOP separately in CDCA2-low and CDCA2-high patients, we should see a comparison of outcomes for CDCA2-low versus CDCA2-high patients, separately for R-CHOP and RB-CHOP treatment groups.
- When xenograft tumors were analyzed by immunohistochemistry (IHC), increased expression of Ki-67 was observed in CDCA2-KO tumors in comparison to SCR controls. This is hard to understand, since Ki-67 is considered to be a marker of proliferating cells, but CDCA2-KO cells proliferated more slowly, both in vitro and in vivo. Furthermore, Ki-67 protein expression is maximal in G2 phase and mitosis (PMID: 34183782), but CDCA2-KO cells have a higher proportion of G1-phase cells.
2) It needs to be stated or shown what is the cell line that was used to compare CDCA2-KO and SCR cells for xenograft experiments. That information is missing for panels E-G of Figure 3, the main text, the Methods, and Supplemental methods.
Author Response
Comment 1: In the REMoDL-B trial, it was previously concluded that ABC and molecular high-grade DLBCL patients benefited from bortezomib, i.e., their outcomes were better with RB-CHOP than with R-CHOP. This study found that DLBCL patients with high expression of CDCA2 had superior survival outcome when treated with RB-CHOP in comparison to R-CHOP, whereas no difference in outcome was observed for patients with low CDCA2. From these two observations, one would expect that CDCA2 is more highly expressed in ABC-DLBCL, at least for a subset of patients, than in GCB-DLBCL. However, it was found that CDCA2 was expressed more highly in GCB-DLBCL cases and in MYC/BCL2 double-expressor patients, which more commonly belong to the GCB group. What is the explanation for these apparent discrepancies?
Response 1: Thank you for addressing this relevant point. Based on our data, CDCA2 mRNA expression is not an unambiguous marker distinguishing ABC- and GCB-classified patients. In fact, there is large spread of CDCA2 expression within each group (Figure 1B). Thus, some ABC-DLBCL patients express higher levels than some of the GCB-DLBCL patients.
Moreover, Sha et al. found that MHG-DLBCL patients have better outcome when treated with RB-CHOP, and 90% of the MHG-classified patients are of the GCB subtype (PMID: 30523719). Unfortunately, we have not information on MHG-subclass or access to the classifier.
We chose to do multiple cox regression, revealing that CDCA2 mRNA expression display prognostic index independent of ABC/GCB and MYC/BCL2. Therefore, we did not include a theoretical speculation of why the expression is higher in GCB-DLBCL patients.
Comment 2: Starting on line 98 of the Results, it is stated that “CDCA2 displayed prognostic impact independently of IPI and ABC/GCB in R-CHOP treated patients whereas no prognostic significance was observed in RB-CHOP treated patients (Supplementary table 3A+B), suggesting CDCA2 as marker of bortezomib response.” Although cell line experiments showed that CDCA2-KO reduced sensitivity to bortezomib, how can CDCA2 be a marker of bortezomib response if no prognostic significance was observed in RB-CHOP treated patients? Perhaps what is needed is a change to Figure 2. Instead of comparing R-CHOP to RB-CHOP separately in CDCA2-low and CDCA2-high patients, we should see a comparison of outcomes for CDCA2-low versus CDCA2-high patients, separately for R-CHOP and RB-CHOP treatment groups.
Response 2: Thank you for pointing this out. To make it clearer, we have added a C and D panel to figure 2, showing R-CHOP and RB-CHOP separately with comparison of CDCA2-low vs. CDCA2-high. The reason why we suggest CDCA2 to be a marker is of bortezomib is that the inferior prognostic outcome observed for R-CHOP treated patients with high CDCA2 expression (panel C) is lost when adding bortezomib to the R-CHOP regimen (panel D). Thereby patients with high CDCA2 levels could benefit for addition of bortezomib to the standard treatment.
Comment 3: When xenograft tumors were analyzed by immunohistochemistry (IHC), increased expression of Ki-67 was observed in CDCA2-KO tumors in comparison to SCR controls. This is hard to understand, since Ki-67 is considered to be a marker of proliferating cells, but CDCA2-KO cells proliferated more slowly, both in vitro and in vivo. Furthermore, Ki-67 protein expression is maximal in G2 phase and mitosis (PMID: 34183782), but CDCA2-KO cells have a higher proportion of G1-phase cells.
Response 3: We agree. However, we believe that the explanation for this can be genetic redundancy. CDCA2 and KI67 regulate protein phosphatase 1 in similar manner and form mitotic exit phosphates by recruiting PP1 (PMID: 27572260), reasoning genetic redundancy. Thus, despite the CDCA2-KO cells proliferate slower, they still manifest as tumors, which could be driven by upregulation of KI67. Genetic redundancy between CDCA2 and KI67 is discussed at page 8, line 237-244.
Comment 4: It needs to be stated or shown what is the cell line that was used to compare CDCA2-KO and SCR cells for xenograft experiments. That information is missing for panels E-G of Figure 3, the main text, the Methods, and Supplemental methods.
Response 4: Thank you for pointing this out. The information has been added to the method section page 10, line 318, the legend of figure 3 page 5, line 149, and to the supplemental methods.